# Cultivar Discrimination of Single Alfalfa (*Medicago sativa* L.) Seed via Multispectral Imaging Combined with Multivariate Analysis

**DOI:** 10.3390/s20226575

**Published:** 2020-11-18

**Authors:** Lingjie Yang, Zuxin Zhang, Xiaowen Hu

**Affiliations:** 1State Key Laboratory of Grassland Agro-ecosystems, College of Pastoral Agriculture Science and Technology, Lanzhou University, Lanzhou 730000, China; yanglj14@lzu.edu.cn (L.Y.); zhangzx13@lzu.edu.cn (Z.Z.); 2Key Laboratory of Grassland Livestock Industry Innovation, Ministry of Agriculture and Rural Affairs, College of Pastoral Agriculture Science and Technology, Lanzhou University, Lanzhou 730000, China; 3Engineering Research Center of Grassland Industry, Ministry of Education, College of Pastoral Agriculture Science and Technology, Lanzhou University, Lanzhou 730000, China

**Keywords:** alfalfa, multispectral imaging, multivariate analysis

## Abstract

Rapid and accurate discrimination of alfalfa cultivars is crucial for producers, consumers, and market regulators. However, the conventional routine of alfalfa cultivars discrimination is time-consuming and labor-intensive. In this study, the potential of a new method was evaluated that used multispectral imaging combined with object-wise multivariate image analysis to distinguish alfalfa cultivars with a single seed. Three multivariate analysis methods including principal component analysis (PCA), linear discrimination analysis (LDA), and support vector machines (SVM) were applied to distinguish seeds of 12 alfalfa cultivars based on their morphological and spectral traits. The results showed that the combination of morphological features and spectral data could provide an exceedingly concise process to classify alfalfa seeds of different cultivars with multivariate analysis, while it failed to make the classification with only seed morphological features. Seed classification accuracy of the testing sets was 91.53% for LDA, and 93.47% for SVM. Thus, multispectral imaging combined with multivariate analysis could provide a simple, robust and nondestructive method to distinguish alfalfa seed cultivars.

## 1. Introduction

Alfalfa (*Medicago sativa* L.), a perennial legume species, is one of the most important crops in semi-arid and arid areas due to its contribution to animal production and cultivated pastures [1]. Alfalfa not only has enormous value as a livestock feed, but it also plays an essential role in reducing soil erosion and nutrient loss, enhancing soil carbon sequestration, and increasing soil nitrogen fertility [2]. Therefore, alfalfa has been an important component of sustainable agricultural systems for many years [3,4].

With the development of breeding techniques, many cultivars of alfalfa have been brought into the market [5]. Different cultivars of alfalfa vary in growth performance, nutrition characteristics, and stress tolerance. Proper cultivars usually show better adaptation to local environment and growth conditions, thus appropriate use of certified seeds plays a vital role in quality and quantity guarantee of alfalfa production. Therefore, guaranteeing the purity of alfalfa seeds with effective cultivar discrimination and sorting is increasingly vital for not only the generated profit of farmers but also the healthy development of the seed industry.

Traditional methods of alfalfa cultivar identification usually rely on the morphological description of plant cultivars in the field [5]. According to the International Union for the Protection of New Varieties of Plants (UPOV), the assessments of distinctness, uniformity, and stability (DUS) throughout the entire growing period in field conditions are used to determine whether a cultivar is distinguishable from another. However, DUS assessments are often costly, time-consuming and restricted to a relatively small number of traits that are influenced by environmental conditions. Molecular testing has been considered as a more efficient way to make cultivar identification with high accuracy [6,7,8]. However, molecular methods are generally required destructive sampling, and not suitable for online measurements and sorting. Thus, it would be in great request to develop a non-destructive, simple, and rapid method for alfalfa cultivars classification.

Multispectral imaging technology integrates conventional imaging and spectroscopy information simultaneously, and is non-destructive, simple, rapid, and does not have the requirement of sample pre-treatment [9,10]. These merits make it suitable for seed purity testing and seed contamination determination. Therefore, this technique has been increasingly applied to compare seeds among species and cultivars. For example, multispectral imaging has been successfully applied to discriminate transgenic and non-transgenic rice seeds [11], to detect differences between rice seeds of different cultivars [12], and to classify maize kernels [13]. However, there is no report on the application of multispectral imaging techniques for rapid discrimination for alfalfa cultivars. Here, we aimed to develop a non-destructive, rapid and high-throughput alfalfa cultivars discrimination method via single seed, based on multispectral imaging techniques in combination with multivariate analysis.

## 2. Materials and Methods

### 2.1. Seed Sample

Twelve alfalfa cultivars as Abi700, Boja, Maverick, Ranger, Sutter, uc-1465, Fado, Vernal, Zhongmu1, Zhongmu3, Dongmu1 and Zhonglan2 were provided by the Germplasm Bank of Cold and Arid Region, Gansu, China. The seeds were kept in water-proof bags in a storage room with an average temperature of −18 °C until used for imaging in January 2020. Photo of the seeds is displayed in Figure 1.

For each cultivar, 200 seeds were used for classification experiments. Origin of each cultivar is listed in Table 1. For each seed lot, 70% were randomly selected as a training set, and 30% were used for testing set.

### 2.2. Multispectral Imaging System

Multispectral images were captured with a VideometerLab4 (Videometer, Hørsholm, Denmark) multispectral imaging system. The images had a high spatial resolution of approximately 40 μm/pixel, consisted of 2192 × 2192 pixels. Before capturing multispectral images, the system was fully calibrated radiometrically and geometrically by using three successive plates: white, dark and a dotted one followed by a light setup calibration. Samples were illuminated by high power light-emitting diodes (LEDs) at the rim of the sphere, ranging from ultraviolet to near-infrared at 19 specific wavelengths: 365, 405, 430, 450, 470, 490, 515, 540, 570, 590, 630, 645, 660, 690, 780, 850, 880, 890, and 970 nm. The LEDs strobed successively in a scan time of approximately five seconds, resulting in a monochrome image at each wavelength at 19 different wavelengths.

### 2.3. Multispectral Image Analysis

Image segmentation was performed using the VideometerLab software version 3.10. Removing the Petri dish and surrounding background preserves the main objects as seeds. Then, attributes of seeds, such as morphological traits and main spectral features of all individual seeds were extracted from the image analysis and processed. The morphological traits included area, length, width, perimeter, diameter area, average edge distance, width/length ratio, compactness circle, compactness ellipse, bounding box side regularity, BetaShape_ a, BetaShape_ b, eccentricity, pointness, width of blob end, CIELab L*, CIE Lab a*, CIE Lab b*, saturation, and hue [9,14]. Based on the reaction of human eye to RGB, Commission Internationale de l’Eclairage (CIE) defined some device-independent color systems such as CIE XYZ, CIE LUV and CIE L*a*b*. CIE L*a*b* system is an improved version of CIE XYZ, which is also an important international standard for measuring color-reproduction errors and simplified mathematical approximation, to a uniform color space composed of perceived color difference, making up for the deficiency of RGB and CMYK color system [15,16]. Parameter L* represents lightness, a* for green-red color and b* for blue-yellow. The color of lightness L* is arranged along the rectangular coordinates a* and b*. Explanation of morphological traits was listed in Appendix A. The extracted spectral signatures of seeds represent the mean intensity of reflected light for every single wavelength calculated from all seed pixels in the image.

### 2.4. Multivariate Data Analysis

Multivariate analysis, including principal component analysis (PCA), linear discrimination analysis (LDA), and support vector machines (SVM), was conducted using *FactoMineR*, *MASS*, and *e1071* packages respectively, in *R*, to classify and distinguish alfalfa cultivar seeds.

#### 2.4.1. PCA

Principal component analysis (PCA) was carried out to identify the patterns hidden in extracted morphological features and spectral data of all seeds, as an explorative multivariate data analysis technique. PCA is commonly used to get an overview of systematic variation in data or as a method for qualitative analysis, it could calculate each principal component contribution and the cumulative score. We used it to explore the possibility of grouping seeds with similar morphology and spectral profiles [17,18,19].

#### 2.4.2. LDA

Linear discriminant analysis (LDA) was used to establish a linear discriminant function that maximizes the ratio between class and within-class variances [18]. In this study, seeds were randomly selected as training (70% of the total samples) and testing sets (remaining 30%). The models for LDA classification were built using the training packages, and the obtained models were validated using independent validation, which was not included in the model construction [20].

For each discriminant model, the recognition levels to new samples in testing sets were defined as the proportion of samples correctly identified to the total number of the seeds in testing sets. It was calculated using the following equation.
Accuracy (%) = Correctly classified seeds/Total number of seeds × 100 (1)

#### 2.4.3. SVM

The least squares-support vector machine (LS-SVM) was proposed by Cortes and Vapnik [21], which was a supervised learning algorithm. As for multivariate function estimation or non-linear classification tasks, SVM performed effectively. Different from other analysis methods, SVM can use fewer training variables or samples in high-dimensional characteristic space. Details of the LS-SVM algorithm can be found in previous research [22,23].

The “Accuracy” discrimination was determined according to Equation (1).

## 3. Results

### 3.1. Morphologic Features of Medicago sativa L. Cultivars Seeds

As for binary features, seeds of uc-1465 are the biggest in size, while those of Zhonglan2 are the smallest. Additionally, uc-1465 showed significant differences in terms of the mean value of area, width, diameter area and average edge distance from other cultivars. However, no significant difference was observed among Sutter, Fado, Vernal, Zhongmu1, and Zhongmu3 in binary features (Appendix A).

For color features, 12 cultivars were different, especially in Fado, whose mean value of CIELab L* was the highest, and the values of CIELab b*, saturation and hue were the lowest. On the contrary, Abi700 had the lowest value of CIELab L*, and the highest value of CIELab b*, saturation, and hue. Zhonglan2 and uc-1465 showed no difference in CIELab a*, CIELab b* saturation and hue. Sutter and Zhongmu1 also showed no difference in CIELab L*, CIELab b* saturation and hue (Appendix A).

For shape features, Fado had the lowest value of compactness circle, BetaShape_ a, BetaShape_ b and pointness, and had the highest value of width/length ratio and eccentricity. Uc-1465 had the highest value of width/length ratio, compactness circle, compactness ellipse, bounding box side regularity, and width of blob end. Seeds of 12 cultivars were almost the same in compactness ellipse, BetaShape_ a, BetaShape_ b, and pointness. Boja, Ranger, and Sutter were similar in width/length ratio, compactness circle, and eccentricity. Zhongmu1, Zhongmu3, Maverick, and Dongmu1 also had the same trend (Appendix A).

When we applied PCA to morphological features, the first three principles components explained 67.41% of the original variance among seeds with 35.40, 20.28 and 11.73% for PC1, PC2 and PC3, respectively (Figure 2a). For spectral features, the explained variance rate for the first three principal components was 60.56, 25.63 and 9.33% of the total variance, respectively (Figure 2b). Moreover, the PCA results based on morphological features and seed spectra also showed that the first three principles components explained 65.98% of the original variance among seeds with 31.60, 19.59 and 14.79% for PC1, PC2, and PC3, respectively (Figure 2c). However, either the score plot of PCA based on morphological, spectral data or their combination failed to separate seeds of different cultivars into 12 distinct groups.

### 3.2. Spectroscopic Analysis

In general, the reflectance of seeds of 12 cultivars showed a similar trend in which the longer the wavelength, the higher the reflectance (Figure 3). However, the reflectance differed significantly among cultivars at each wavelength. For example, Fado had a significantly higher reflectance value in the spectral range from 365 to 780 nm than that of other cultivars, while showed a lower reflectance value in 850 to 970 nm compared to other cultivars in addition to Zhongmu1, Zhongmu3, Vernal, Dongmu 1 and Zhonglan2 (Appendix A). Seeds of 12 cultivars can be divided into 3 groups according to their light reflectance on 365 to 540 nm and 880 to 970 nm. Group one only includes Fado, Group two including Zhongmu1, Zhongmu3, Dongmu1 and Zhonglan2, and the group three including Abi 700, Boja, Marcrick, Ranger, Sutter uc-1465 and Vernal. Regarding uc-1465, Boja and Maverick, they showed a very similar reflectance pattern which has the lowest value from 365 to 540 nm, and the highest value from 780 to 970 nm.

### 3.3. Discrimination Models for Seed Classification

As explained before, two multivariate discriminate analysis models were developed on the basis of different data sources: morphological features data, spectral features data and a combination of morphological and spectral features data. The results revealed that the LDA model based on morphological features data had a classification accuracy of 43.63% and 42.22% for training and independent testing datasets, respectively (Appendix A). On the other hand, Fado presented a distinct distance from the other cultivars, the other eleven cultivars could not be separated (Figure 4a). In contrast, the accuracy of the LDA model was greatly improved on the basis of the spectral data, which exhibited a high discrimination accuracy up to 87.50% and 86.81% with training and testing datasets, respectively (Appendix A) (Figure 4b). When using the combined morphological features and spectral data, the overall correct classification ratios in training and testing sets were 91.96%, 92.44% and 91.53%, respectively (Appendix A). However, the classification accuracy differs greatly among cultivars. For example, the classification accuracy for Fado is 100%, while that of Zhongmu3 is as low as to 11.67% with LDA.

For the LDA model with morphology data, the first five features explained 72.98% of the total variation, followed by width/length ratio (24.60%), diameter area (15.56%), compactness ellipse (15.33%), eccentricity (10.13%), and width (7.36%), suggesting that morphological discrimination between cultivars is mainly based on shape features (Figure 5a). For spectral data, 470 nm (12.75%), 490 nm (9.74%), 940 nm (8.79%), 430 nm (7.44%), 970 nm (7.28%) were first five features add up to 46.00% variation for LDA (Figure 5c). For the combination of morphology and spectral data (Figure 5e), width/length ratio (15.53%), diameter area (11.60%), compactness ellipse (8.81%), eccentricity (5.76%), and width (5.14%) were the first five features and accounted for 46.86% variation for LDA, showing that morphological features had a great contribution for cultivar discrimination.

The classification accuracy of the SVM model based on morphological data was 45.95% and 45.14% for training and independent testing datasets, respectively (Table 2). When based on spectral data, the accuracy was 89.46% and 87.78% for training and testing, respectively (Table 3). Using the combination of morphological features and spectral data, the percentage of overall correct classification ratio in training and validation sets was 95.65% and 93.47%, respectively (Table 4).

For the SVM model with morphology data (Figure 5b), the first five features, CIELab L* (33.76%), saturation (20.08%), CIELab b* (18.76%), CIELab a* (6.79%) and average edge distance (4.82%), accounted for 84.21% variation of the model, indicated seed color played an important role in cultivar discrimination. For spectral features (Figure 5d), 970 nm (12.77%), 940 nm (12.27%), 880 nm (10.48%), 850 nm (9.66%), 780 nm (7.45%) were first five features add up to 52.63% variation for SVM. For the combination of morphology and spectral data (Figure 5f), 970 nm (9.67%), 940 nm (9.29%), CIELab L* (8.19%), 880 nm (7.94%) and 850 nm (7.32%) were the first five features, accounting for 42.41% variation of SVM, suggesting that spectral and color features were more important for SVM classification.

## 4. Discussion and Conclusions

Although the morphological features clearly show that seeds of different alfalfa cultivars vary in binary, shape, and color in terms of mean value, the large variation within cultivar overrides the difference among cultivars. Thus, using any single morphological feature would not distinguish cultivars successfully.

The difference between seed spectrums reflected the difference in the seed coat texture and chemical composition which may differ among different cultivars [24]. The reflectance at special wavelength differed significantly among cultivars in wheat [25] and corn [26]. Consistent with this, our study shows that cultivars can be divided into three groups according to spectral reflectance, these three groups partly reflect the origin of cultivars. For example, the four cultivars of group two are all from China, and the cultivars of group three are all from America. The reason is possibly that cultivars originated from the same place share similar growth conditions as well as seed processing which play a key role in shaping the morphological features and chemical composition of seeds. More importantly, cultivars originated from the same place may also share the same breeding material, and thus have a similar genetic background. For example, Zhongmu1 and Zhongmu3 have a close relationship during breeding. Further, although reflectance differed significantly in cultivars in terms of mean value at each wavelength, we still can find a heavy overlap due to large variation within cultivars, suggesting any single spectral trait is not practical for cultivar discrimination.

As an unsupervised discrimination method, principal component analysis (PCA) is generally used to get an overview of systematic variation in data, showing the sample distribution pattern. A previous study [27] showed that PCA can be used to discriminate maize seeds of different varieties via hyperspectral imaging. Additionally, a distinct difference via multispectral imaging has been observed from conventional and glyphosate-resistant soybean seeds to their hybrid descendants based on PCA scatter plot [28]. In contrast with these, although there are some differences among cultivars in both morphological and spectral traits, the PCA scatter plot fails to distinguish cultivars in our study. A possible reason is that *Medicago sativa* is a cross-pollination plant, and variation within the group is relatively large. Thus, PCA would not detect the difference among cultivars when variation within the group is close to the variation among groups, since PCA is aimed to maximize the variation of the samples. Moreover, the first three principal components account for less than 65% of total variance whatever the data sources used in the present study. Thus, the loss of information may further lead to a failure to separate cultivars by PCA.

The average accuracy of cultivars classification using the LDA and SVM model on the basis of morphological features is also very low, suggesting that seeds of different cultivars have similar morphological features. Thus, using either a single morphological feature or their combination could not distinguish seed cultivars regardless of the discrimination model. However, for cultivars such as Fado, accuracy is 100% either using LDA or SVM, and this result suggests that it is practical to discriminate some special cultivars, such as Fado, with LDA or SVM models based on morphological traits. The relative importance histogram shows that CIE L*, CIE a* and CIE b* play an important role in the SVM model. Consistent with this, Zhang and Lu [29] found that CIE L*a*b* combined with SVM was effective for the detection of browning degree in mangoes.

In contrast, the average accuracy of cultivar classification was greatly improved when using spectral data with a value of 86.81% and 87.78% for LDA and SVM, respectively. An interesting thing is that the contribution of reflectance at various wavelengths proves to largely depend on the discrimination model. Reflectance from 430 nm to 490 nm in the visible range has a high contribution to the LDA model, while the wavelength from 780 nm to 970 nm in the NIR range is particularly important for the SVM model. A possible reason is that the visible range detecting the color of seeds as wavelength from 450 nm to 470 nm is sensitive to chlorophyll, and NIR reflects the chemical difference, as 600 nm to 950 nm contains a wealth of useful information related to fats and proteins [12].

Furthermore, when the combination of morphological and spectral data is used, LDA and SVM have an average accuracy as high as 91.53% and 93.47% in alfalfa cultivars discrimination, respectively. The classification accuracy of different cultivars is ranged from 80.00% to 100% both for LDA and SVM. In particular, the classification accuracy of 10 cultivars is higher than 90.00% with SVM, implying multispectral imaging together with multivariate analysis could be a promising technique to identify alfalfa cultivar with high efficiency. It is also worth noting that, for some closely related cultivars, such as Ranger and Abi700, they are cross-misclassified with both LDA and SVM model, thus leading to relatively low classification accuracy. As we discussed above, cultivars originating from the same place may have similar growth conditions, seed processing as well as similar genetic background, thus resulting in similar seed morphology and chemical compositions.

In brief, our study clearly shows that seeds of different alfalfa cultivars differ in morphology and spectral traits, which exhibit a large variation both within and among cultivars. It is difficult to discriminate seeds of alfalfa with different cultivars on the basis of any single seed trait or unsupervised discrimination method such as PCA. However, supervised discrimination methods such as LDA and SVM have a high accuracy of cultivar classification based on morphological and spectral data, suggesting that multispectral imaging analysis together with multivariate analysis could be an efficient way to alfalfa cultivar discrimination.

## Figures and Tables

**Figure 1 sensors-20-06575-f001:**
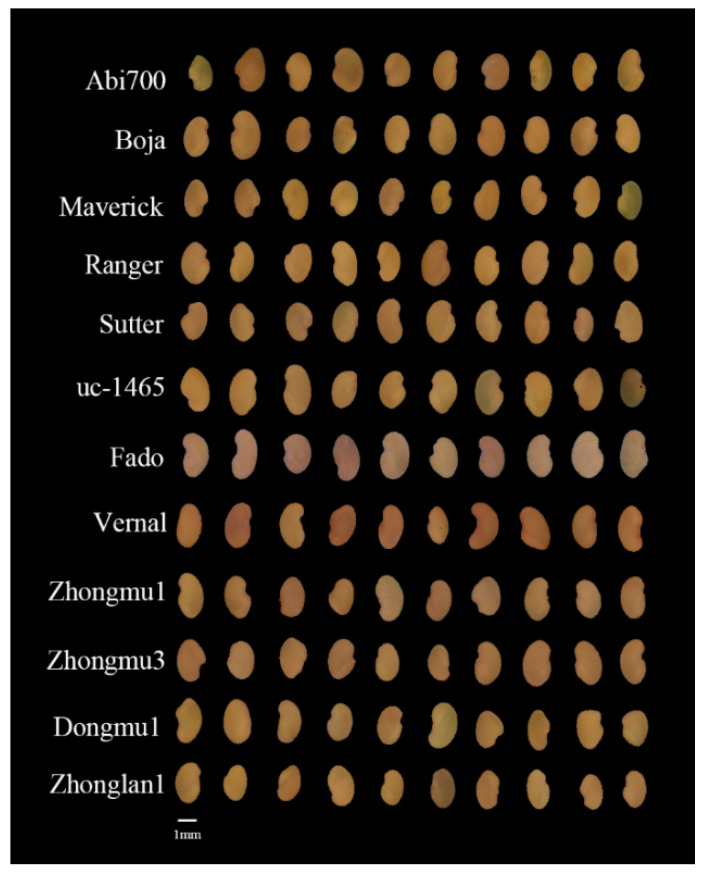
Seed image of 12 *Medicago sativa* L. cultivars.

**Figure 2 sensors-20-06575-f002:**
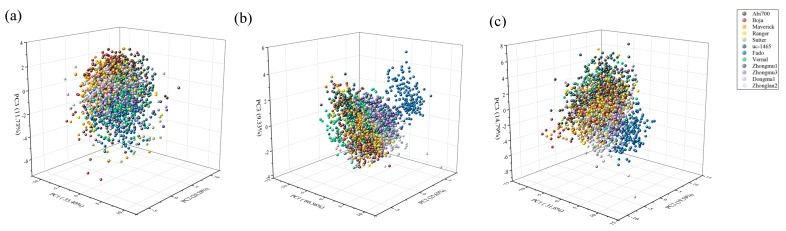
Three-dimensional plot of the first three principal components (PCs) for (**a**) morphological, (**b**) spectral and (**c**) morphological combined with spectral features dataset in 12 cultivars.

**Figure 3 sensors-20-06575-f003:**
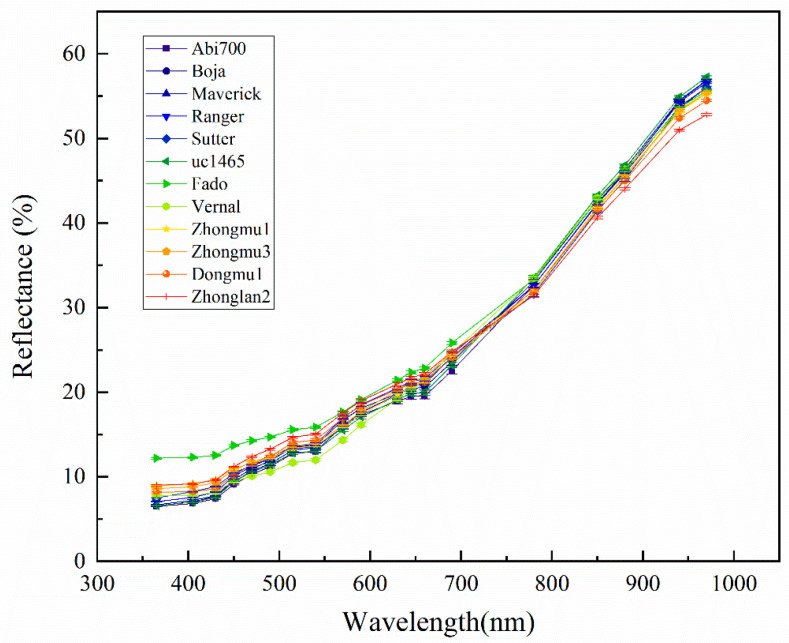
Mean light reflectance of 19 wavelengths (nm) in 12 *Medicago sativa* L. cultivars.

**Figure 4 sensors-20-06575-f004:**
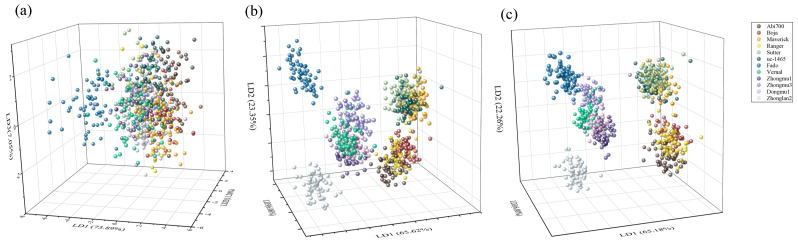
Score plot of linear discrimination analysis (LDA) model for discrimination twelve cultivars seeds of *Medicago sativa* L. based on (**a**) morphological, (**b**) spectral and (**c**) morphological combined with spectral features.

**Figure 5 sensors-20-06575-f005:**
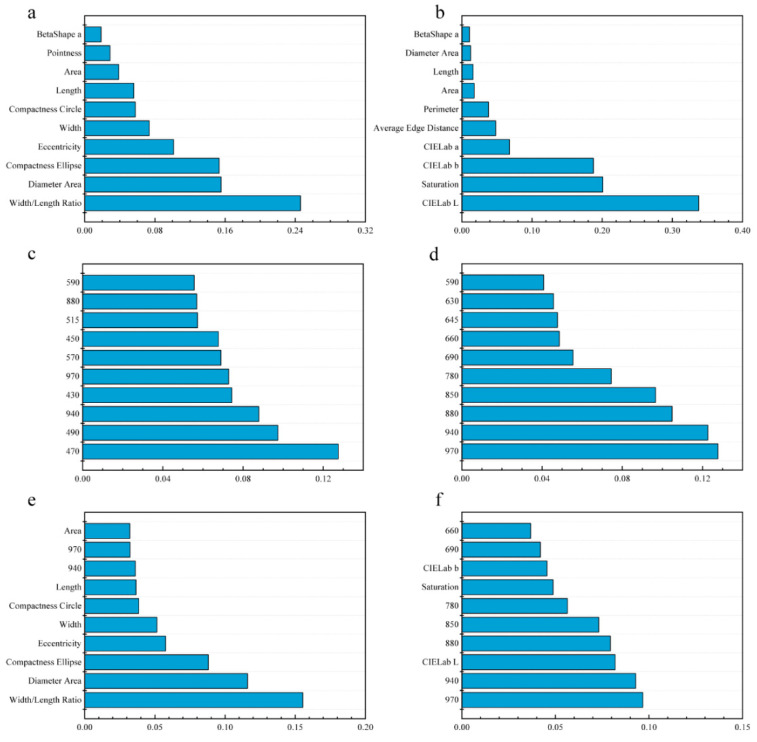
Relative importance of morphological, spectral and morphological combined with spectral features for LDA and support vector machine (SVM) model. (**a**) morphological features of LDA. (**b**) morphological features of SVM. (**c**) spectral features of LDA. (**d**) spectral features of SVM. (**e**) morphological combined with spectral features of LDA. (**f**) morphological combined with spectral features of SVM.

**Table 1 sensors-20-06575-t001:** Seeds information of *Medicago sativa* L.

Number	Latin Name	Origin	Cultivar
1	*Medicago sativa* L.	United States	Abi700
2	*Medicago sativa* L.	United States	Boja
3	*Medicago sativa* L.	United States	Maverick
4	*Medicago sativa* L.	United States	Ranger
5	*Medicago sativa* L.	United States	Sutter
6	*Medicago sativa* L.	United States	uc-1465
7	*Medicago sativa* L.	France	Fado
8	*Medicago sativa* L.	United States	Vernal
9	*Medicago sativa* L.	China	Zhongmu1
10	*Medicago sativa* L.	China	Zhongmu3
11	*Medicago sativa* L.	China	Dongmu1
12	*Medicago sativa* L.	China	Zhonglan2

**Table 2 sensors-20-06575-t002:** Discrimination performance based on SVM with morphology features of 12 *Medicago sativa* L. cultivars.

	Predict	Actual												Total (%)
		Abi700	Boja	Maverick	Ranger	Sutter	uc-1465	Fado	Vernal	Zhongmu1	Zhongmu3	Dongmu1	Zhonglan2	
Training	Abi700	83	22	23	13	1	14	0	2	0	2	10	4	
(*n* = 140)	Boja	22	55	26	18	1	4	0	0	3	2	9	8	
	Maverick	8	25	48	19	2	5	0	3	3	2	7	4	
	Ranger	6	17	14	25	4	8	0	6	3	8	10	5	
	Sutter	2	0	1	6	54	7	0	5	20	19	12	11	
	uc-1465	10	3	6	12	11	79	0	5	11	16	19	4	
	Fado	0	0	0	0	0	0	140	0	1	0	0	1	
	Vernal	3	5	7	12	10	5	0	93	26	21	2	2	
	Zhongmu1	1	0	1	5	19	3	0	16	50	29	7	4	
	Zhongmu3	0	3	3	8	9	7	0	4	9	25	2	2	
	Dongmu1	4	5	5	12	11	6	0	0	4	5	42	17	
	Zhonglan2	1	5	6	10	18	2	0	6	10	11	20	78	
	Accuracy (%)	59.29	39.29	34.29	17.86	38.57	56.43	100.00	66.43	35.71	17.86	30.00	55.71	45.95
Testing	Abi700	30	11	9	5	0	7	0	0	0	0	6	1	
(*n* = 60)	Boja	13	24	12	5	1	1	0	1	1	0	3	3	
	Maverick	1	9	21	10	0	1	0	4	1	0	4	1	
	Ranger	4	5	8	17	2	4	0	1	1	2	3	0	
	Sutter	2	0	0	2	17	2	0	2	7	13	3	3	
	uc-1465	6	2	0	2	3	36	0	2	3	4	6	1	
	Fado	0	0	0	0	0	0	59	0	2	0	0	0	
	Vernal	1	4	5	4	6	1	0	38	6	12	1	0	
	Zhongmu1	0	0	0	3	9	0	1	6	25	13	3	4	
	Zhongmu3	0	0	0	3	6	4	0	5	5	7	2	1	
	Dongmu1	2	1	2	5	6	1	0	0	2	3	14	9	
	Zhonglan2	1	4	3	4	10	3	0	1	7	6	15	37	
	Accuracy (%)	50.00	40.00	35.00	28.33	28.33	60.00	98.33	63.33	41.67	11.67	23.33	61.67	45.14

**Table 3 sensors-20-06575-t003:** Discrimination performance based on SVM with spectral of 12 *Medicago sativa* L. cultivars.

	Predict	Actual												Total (%)
		Abi700	Boja	Maverick	Ranger	Sutter	uc-1465	Fado	Vernal	Zhongmu1	Zhongmu3	Dongmu1	Zhonglan2	
Training	Abi700	122	5	0	18	0	0	0	0	0	0	0	0	
(*n* = 140)	Boja	5	113	0	19	1	4	0	0	0	0	0	0	
	Maverick	0	1	119	1	4	14	0	0	0	0	0	0	
	Ranger	13	21	0	102	0	0	0	0	0	0	0	0	
	Sutter	0	0	7	0	133	4	0	0	0	0	0	0	
	uc-1465	0	0	14	0	2	118	0	0	0	0	0	0	
	Fado	0	0	0	0	0	0	140	0	0	0	0	0	
	Vernal	0	0	0	0	0	0	0	132	3	2	0	0	
	Zhongmu1	0	0	0	0	0	0	0	1	123	1	11	0	
	Zhongmu3	0	0	0	0	0	0	0	6	0	136	4	0	
	Dongmu1	0	0	0	0	0	0	0	1	14	1	125	0	
	Zhonglan2	0	0	0	0	0	0	0	0	0	0	0	140	
	Accuracy (%)	87.14	80.71	85.00	72.86	95.00	84.29	100.00	94.29	87.86	97.14	89.29	100.00	89.46
Testing	Abi700	48	2	0	9	0	0	0	0	0	0	0	0	
(*n* = 60)	Boja	3	51	0	9	0	2	0	0	0	0	0	0	
	Maverick	0	0	52	0	1	3	0	0	0	0	0	0	
	Ranger	8	7	0	42	0	1	0	0	1	0	0	0	
	Sutter	0	0	1	0	59	3	0	0	0	0	0	0	
	uc-1465	1	0	7	0	0	51	0	0	0	0	0	0	
	Fado	0	0	0	0	0	0	60	0	0	0	0	0	
	Vernal	0	0	0	0	0	0	0	57	4	0	0	0	
	Zhongmu1	0	0	0	0	0	0	0	1	45	2	7	0	
	Zhongmu3	0	0	0	0	0	0	0	1	0	57	2	0	
	Dongmu1	0	0	0	0	0	0	0	1	10	1	51	1	
	Zhonglan2	0	0	0	0	0	0	0	0	0	0	0	59	
	Accuracy (%)	80.00	85.00	86.67	70.00	98.33	85.00	100.00	95.00	75.00	95.00	85.00	98.33	87.78

**Table 4 sensors-20-06575-t004:** Discrimination performance based on SVM with morphology and spectral features of 12 *Medicago sativa* L. cultivars.

	Predict	Actual												Total (%)
		Abi700	Boja	Maverick	Ranger	Sutter	uc-1465	Fado	Vernal	Zhongmu1	Zhongmu3	Dongmu1	Zhonglan2	
Training	Abi700	127	4	0	6	0	0	0	0	0	0	0	0	
(*n* = 140)	Boja	3	131	1	2	0	0	0	0	0	0	0	0	
	Maverick	0	0	135	1	1	9	0	0	0	0	0	0	
	Ranger	10	5	1	131	0	0	0	0	0	0	0	0	
	Sutter	0	0	0	0	139	0	0	0	0	0	0	0	
	uc-1465	0	0	3	0	0	131	0	0	0	0	0	0	
	Fado	0	0	0	0	0	0	140	0	0	0	0	0	
	Vernal	0	0	0	0	0	0	0	140	0	1	0	0	
	Zhongmu1	0	0	0	0	0	0	0	0	129	0	9	0	
	Zhongmu3	0	0	0	0	0	0	0	0	0	136	3	0	
	Dongmu1	0	0	0	0	0	0	0	0	11	3	128	0	
	Zhonglan2	0	0	0	0	0	0	0	0	0	0	0	140	
	Accuracy (%)	90.71	93.57	96.43	93.57	99.29	93.57	100.00	100.00	92.14	97.14	91.43	100.00	95.65
Testing	Abi700	53	2	0	2	0	0	0	0	0	0	0	0	
(*n* = 60)	Boja	2	55	0	4	0	1	0	0	0	0	0	0	
	Maverick	0	0	55	0	0	8	0	0	0	0	0	0	
	Ranger	5	2	1	54	0	1	0	0	0	0	0	0	
	Sutter	0	0	2	0	59	1	0	0	0	0	0	0	
	uc-1465	0	1	2	0	0	49	0	0	0	0	0	0	
	Fado	0	0	0	0	0	0	60	0	0	0	0	0	
	Vernal	0	0	0	0	0	0	0	60	0	0	1	0	
	Zhongmu1	0	0	0	0	1	0	0	0	55	0	4	1	
	Zhongmu3	0	0	0	0	0	0	0	0	0	60	1	0	
	Dongmu1	0	0	0	0	0	0	0	0	5	0	54	0	
	Zhonglan2	0	0	0	0	0	0	0	0	0	0	0	59	
	Accuracy (%)	88.33	91.67	91.67	90.00	98.33	81.67	100.00	100.00	91.67	100.00	90.00	98.33	93.47

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
