# Peer review of "Cultivar Discrimination of Single Alfalfa (Medicago sativa L.) Seed via Multispectral Imaging Combined with Multivariate Analysis"

_sensors, 2020, doi:10.3390/s20226575_

Round 1

Reviewer 1 Report

I suggest reconsidering this manuscript after a major revision.

I can't agree with the title: Variety identification of single alfalfa (Medicago sativa) seed via multispectral imaging combined with multivariate analysis, because I suppose that taking more cultivars we couldn't identify them. So I would suggest changing 'identification' on comparison or something similar. Moreover, I am geneticists and I am pretty sure that in case of cultivars with huge genetic similarity or with broad intracultivar diversity this method wouldn't work. Medicago sativa should be corrected on Medicago sativa L. And variety or cultivar? Rather cultivar.

I don't understand why in the discussion the authors didn't mention other people papers. Discussion - should be a kind of confrontation of your results with the results of other scientists. Without putting your results in the wider context it is hard to assess the impact of your work on science and decide if your results broaden our knowledge. Moreover, some information from discussion should be moved to Results as they describe the results. The lack of these observations within Results made them uncomplete.

The presentation of results is boring. More sophisticated methods would improve the interest of readers.

Summarising, my main concern is a lack of real discussion and citing less than 20 other authors papers. As well as I am not sure if you are authorised to call all described attempts as a method of variety/cultivar identification.

Reviewer 2 Report

The paper describes several approaches to classifying alfalfa seeds using morphological, or spectral data, or both.  It concludes that LDA or SVM that uses both morphological and spectral information can classify alfalfa seed types reliably.

First, I could not find Figure 6 (line 161).  Associated with this (and perhaps figure 6 showed this) there should be at least one plot of reflectance versus wavelength showing the differences noted in the text between the 3 different group classes one sees with respect to spectral reflectance.

Line 75 has the phrase "...a doted one..." with respect to calibration.  Should this read "...a dotted one..." instead?

Reviewer 3 Report

It is appreciatable that the authors have come up with the idea of differentiating the varieties of an important forage crop like Medicago sativa (alfalfa). The approach is interesting to readers of the journal. However the following concerns must be addressed  

  1. Please define the type of CIE coordinate was used in multispectral image analysis
  2. Please explain the color coordinate system to benefit the audience of the journal – explain each parameter and their significance in CIE color coordinates of L*a*B*
  3. The authors have used the L*a*b* coordinate system of CIE to discriminate the color variation -Is there a reason for using the individual parameters L*, a* and b* instead of the ΔEab because L* is measure of darkness or lightness, a* is a measure of red or green tinge, and b* measures yellow versus blue color compositions.  
  4. The authors may discuss the broader application of CIE coordinate systems in other fields too – a few citations on this line is better - Results in Physics, 15, 2019, 102648; Chemical Communications, 49 (91), 10742
  5. It is clear from the study that the spectral features differentiated the varieties very well. – why didn’t the authors analyze the spectral features alone in accuracy of differentiating the varieties?
  6. The reflectance in each wavelength must be presented as a line plot instead of table – It will be easier to differentiate the features as a plot. – The measurement of only a few wavelength is understood here. However, it makes the manuscript readable
  7. A relative importance plot of each wavelength in spectral feature must be discussed.
  8. Figures 2 and 3 may be moved to the supporting information.
  9. Minor English corrections must be done.

Round 2

Reviewer 1 Report

The current version is much better.

Reviewer 2 Report

The authors have done a convincing job showing that combining morphological and spectral imaging information can improve the classification accuracy of identifying alfalfa seeds with different origins.  This is one of the strengths of imaging spectroscopy.

I appreciate the authors' incorporation of changes from myself and other reviewers, which has improved the paper compared to its original version.

This manuscript is a resubmission of an earlier submission. The following is a list of the peer review reports and author responses from that submission.